# Transcriptome Remodeling in Gradual Development of Inverse Resistance between Paclitaxel and Cisplatin in Ovarian Cancer Cells

**DOI:** 10.3390/ijms21239218

**Published:** 2020-12-03

**Authors:** Jolanta Szenajch, Alicja Szabelska-Beręsewicz, Aleksandra Świercz, Joanna Zyprych-Walczak, Idzi Siatkowski, Michał Góralski, Agnieszka Synowiec, Luiza Handschuh

**Affiliations:** 1Laboratory for Molecular Oncology and Innovative Therapies, Military Institute of Medicine, 04-141 Warsaw, Poland; asynowiec@wim.mil.pl; 2Department of Mathematical and Statistical Methods, Poznań University of Life Sciences, 60-637 Poznań, Poland; alicja.szabelska@up.poznan.pl (A.S.-B.); joanna.zyprych-walczak@up.poznan.pl (J.Z.-W.); idzi.siatkowski@up.poznan.pl (I.S.); 3Laboratory of Genomics, Institute of Bioorganic Chemistry, Polish Academy of Science, 61-704 Poznań, Poland; Aleksandra.Swiercz@cs.put.poznan.pl (A.Ś.); gurral@poczta.onet.pl (M.G.); Luiza.Handschuh@ibch.poznan.pl (L.H.); 4Institute of Computing Science, Poznan University of Technology, 60-965 Poznań, Poland

**Keywords:** ovarian cancer, paclitaxel, cisplatin, inverse resistance, gene expression profile, network analysis, osteogenesis, chondrogenesis, osteomimicry

## Abstract

Resistance to anti-cancer drugs is the main challenge in oncology. In pre-clinical studies, established cancer cell lines are primary tools in deciphering molecular mechanisms of this phenomenon. In this study, we proposed a new, transcriptome-focused approach, utilizing a model of isogenic cancer cell lines with gradually changing resistance. We analyzed trends in gene expression in the aim to find out a scaffold of resistance development process. The ovarian cancer cell line A2780 was treated with stepwise increased concentrations of paclitaxel (PTX) to generate a series of drug resistant sublines. To monitor transcriptome changes we submitted them to mRNA-sequencing, followed by the identification of differentially expressed genes (DEGs), principal component analysis (PCA), and hierarchical clustering. Functional interactions of proteins, encoded by DEGs, were analyzed by building protein-protein interaction (PPI) networks. We obtained human ovarian cancer cell lines with gradually developed resistance to PTX and collateral sensitivity to cisplatin (CDDP) (inverse resistance). In their transcriptomes, we identified two groups of DEGs: (1) With fluctuations in expression in the course of resistance acquiring; and (2) with a consistently changed expression at each stage of resistance development, constituting a scaffold of the process. In the scaffold PPI network, the cell cycle regulator—polo-like kinase 2 (PLK2); proteins belonging to the tumor necrosis factor (TNF) ligand and receptor family, as well as to the ephrin receptor family were found, and moreover, proteins linked to osteo- and chondrogenesis and the nervous system development. Our cellular model of drug resistance allowed for keeping track of trends in gene expression and studying this phenomenon as a process of evolution, reflected by global transcriptome remodeling. This approach enabled us to explore novel candidate genes and surmise that abrogation of the osteomimic phenotype in ovarian cancer cells might occur during the development of inverse resistance between PTX and CDDP.

## 1. Introduction

Drug resistance is a phylogenetically old and universal phenomenon taking place when the disease becomes tolerant to pharmaceutical treatment. Contemporary medicine incessantly faces these challenges, from bacteria acquiring resistance to antibiotics to human drug-resistant cancers. Whatever drug is employed for treatment, the acquirement of resistance is almost inevitable. In this never-ending arms race, continuous research on molecular and cellular mechanisms of resistance in cancers is essential to identify new potential drug targets and refractory/sensitivity biomarkers. Nowadays, when high-throughput techniques of genomics, transcriptomics, proteomics, and metabolomics allow for a simultaneous study of thousands of factors, our possibilities in this field are much greater than 30 years before, when the first cell lines with in vitro induced drug resistance were developed in laboratories. Established cancer cell lines are still the primary tools in these investigations, so there is a constant need for development ones.

Ovarian cancer cells are very popular models in drug resistance research due to the epidemiology and clinical course of this disease. This is the eighth most frequently diagnosed cancer and the eighth cause of cancer death in women worldwide [1]. A standard care of ovarian cancer is based on cytoreductive surgery and chemotherapy with platinum derivatives (carboplatin, cisplatin (CDDP)) alone or combined with taxanes (paclitaxel (PTX), docetaxel) [2]. Although the first-line chemotherapy usually gives 80% positive responses, the majority of patients eventually die due to recurrent disease resistance to the administered drugs [3,4]. Therefore, the acquired resistance remains a major obstacle in ovarian cancer therapy.

The initial mechanisms of CDDP and PTX actions are different. CDDP binds to DNA and causes the formation of adducts, which impede cellular processes that require the separation of both DNA strands, such as replication and transcription [5]. PTX causes hyper stabilization of polymerized microtubules, which results in blocking mitosis at the metaphase/anaphase transition and can delay or prevent cell division [6]. Both drugs eventually trigger apoptosis and at the final stages they may share common signaling pathways and networks [7].

The mechanisms of resistance to these drugs are also different. Well-studied mechanisms conferring resistance to CDDP are as follows:(1)Decreased CDDP influx, mediated by the copper transporter: High-affinity Cu transporter CTR1.(2)Increased CDDP efflux by copper exporters: ATPase copper transporting alpha and beta (ATP7A and ATP7B).(3)CDDP inactivation by glutathione S-transferase (GST) and cytochrome P450 (CYP) systems.(4)Restoration of DNA damage repair by bringing back the BRCA1/2 function [8,9].

PTX induces several resistance mechanisms; two of them which may be targets of therapeutic intervention are as follows:(1)Increased PTX efflux by membrane-bound transporters, particularly by over-expressed proteins from the ABC (ATP-binding cassette) family.(2)Modulation of the mitochondrial apoptotic pathway, which is induced by a prolonged cell cycle arrest at the G2/M check point after the PTX and/or CDDP treatment. Increased expression of anti-apoptotic proteins, such as BCL2 apoptosis regulator (BCL2) and BCL2-like 1 (BCL-X_L_); or decreased expression of pro-apoptotic proteins, such as BCL2 associated X, apoptosis regulator (BAX) and caspases, conferring resistance to both PTX and CDDP [8,9].

Another important aspect of resistance acquirement is the cross-resistance and collateral sensitivity. When cancer cells are treated with one drug in vitro or in vivo, the development of resistance or sensitivity to other drugs is a likely outcome. This occurs as a bystander effect and as for now we are not able to predict (e.g., how CDDP administration will influence the response to PTX and vice versa) and control these processes.

Stordal et al. in their systematic review of the literature analyzed 137 cell models of acquired resistance both in commercial cell lines and cell lines developed from patients before and after chemotherapy. Sixty eight percent of these cell lines were inversely resistant to PTX and CDDP (i.e., exhibited the increased resistance to one drug and decreased or at least unchanged resistance to the other drug); 17% of the analyzed lines were cross-resistant (i.e., exhibited the increased resistance to both drugs); and 14% did not exhibit the increased resistance to any drug. The dominant trend of acquiring the inverse resistance seems to be characteristic for different types of cancers: Ovarian cancer cell lines made 45.3% of the investigated set; small cell lung carcinoma cell lines—21.2% and other cancer types—33.5%. Furthermore, the similar inverse resistance pattern was observed for different pairs of platinum derivatives and taxanes: CDDP vs. docetaxel, carboplatin vs. PTX, and carboplatin vs. docetaxel [10]. Although this trend has been known and widely explored in laboratories with a large number of anti-cancer drugs and different cancer cells for about 40 years, it has not been extensively exploited in the clinical practice. In the clinic, a combination chemotherapy is commonly used, which leads to a simultaneous response to several drugs and hinders the rational selection of collateral sensitivity agent [11,12].

The aim of the present study was to create a model of human ovarian cancer cell lines with gradually developed resistance to PTX and, using a transcriptome focused approach, look at the mechanisms of resistance acquirement as an evolution process. A comprehensive transcriptome analysis revealed widespread changes in gene expression and allowed for the identification of two groups of differentially expressed genes (DEGs): (1) With fluctuations in expression at consecutive stages of the process; (2) with a consistently changed expression at each stage of resistance development—constituting a scaffold of the process. A silico analysis of functional interactions of proteins encoded by those genes indicated that the resistance development involved a cell cycle regulator—polo-like kinase 2 (PLK2) and proteins from the TNF ligand and receptor family, as well as, unexpectedly, proteins associated with the nervous system development and also with osteo- and chondrogenesis. Therefore, we suspect that during the development of inverse resistance between PTX and CDDP, the abrogation of the osteomimic phenotype in ovarian cancers cells may take place. In order to deepen our insight into these processes, these findings need to be expanded by biological and physiological investigations.

Our cellular model was patented and will be available soon to the worldwide scientific community in one of the international cell culture collections.

## 2. Results

### 2.1. Development of the A2780 Cell Model with Acquired Resistance to PTX

The A2780 ovarian cancer cell line was used as a parental cell line to develop a series of drug resistant sublines by continuous growing of A2780 cells in step-wise increases in PTX. We have chosen this way in the aim to develop the so-called “high-level laboratory model” with several- to several dozen-fold increase from the half maximal inhibitory concentration (IC_50_). We expected that our model will be stably resistant and provide a large number of molecular changes for transcriptome analyses [13]. The family tree of generated sublines is presented in Figure 1, whereas Appendix A depicts the morphology differences in selected subline cultures.

The value of IC_50(120h)_ for PTX in A2780 cells was determined as 4.62 nM (Table 1), so the treatment was commenced at 0.25 nM PTX (around 5% of IC_50(120h)_). As seen, low PTX concentrations 0.25–1 nM were very well tolerated and after 7 days of continuous treatment the cells were ready for bearing the next, double dose. Difficulties started at a 2 nM concentration (about 50% of IC_50(120h)_ dose)—only after the 4-day treatment the culture began to die and needed 7 days to recover. In general, cells treated with 2 nM or higher doses of PTX needed rest periods during adaptation to the drug and 30–63 days to develop the stable growth given in the PTX concentration. It is worth mentioning that two PTX concentrations: 4 and 16 nM PTX were overcome most dramatically—the histories of final sublines development: A/4PTX-**63d** (**5** + 14 + **58**) and A/16PTX-**56d** (**26** + 11 + **30**) (Figure 1) show that they needed several days of recovery period and about two months of PTX treatment to reach the stable tolerance and robust growth in these PTX concentrations. In the further part of this text, the full designations of sublines: A/4PTX-63d, A/8PTX-33d, A/16PTX-56d, A/32PTX-48d, A/64PTX-30d, and A/128PTX-37d are used interchangeably with abbreviated names: A/4PTX, A/8PTX, A/16PTX, A/32PTX, A/64PTX, and A/128PTX, respectively.

In summary, 10 consecutive treatment cycles with a series of doubling PTX doses: 0.25–0.5–1–2–4–8–16–32–64–128 nM were performed on A2780 cell cultures over a 17-month period. According to the literature, very low doses of PTX (less than IC_50_) result in a slight inhibition of cell growth in vitro, but do not induce sustained mitotic arrest. Low or clinically relevant doses of 5–200 nM cause mitotic arrest and cell death [14]. In ovarian cancer patients, after the bolus infusion of PTX (175 mg/m^2^ in 3 h) its concentration in plasma after the end of infusion achieved 860 nM in 2 h, 57 nM in 24 h, and fell to 21 nM in 48 h [15]. Data, concerning the concentration of PTX in the tumor, have not been published yet. In our study performed on cells cultured in a single layer, it seems to be reasonable that PTX concentrations from the lower end and even more decreased would better mimic in the vivo exposure, since in vitro growing cells have an easier access to the drug than the in vivo growing tumor. Taking into consideration all these aspects, we recognized that 4, 8, 16, 32, 64, and 128 nM PTX concentrations, used for resistance development, may be reasonably selected for further studies.

### 2.2. Comparison of Cell Subline Series with a Very Wide Range of Induced Resistance to PTX and Sensitivity to CDDP

#### 2.2.1. Determination of Drug Concentrations for Selected Values of Viability

Since it is widely known that a treatment with one anti-cancer drug results very often in acquiring resistance not only to this drug, but also in collaterally developing resistance or sensitivity to other anti-cancer drugs, we decided to check the resistance of the generated series of A2780 sublines to both PTX and CDDP as two basal chemotherapeutics used in ovarian cancer therapy.

To estimate the range of resistance we performed the cytotoxicity assay. The preliminary results from the first attempt provided the rough idea about the profile of generated resistance:A2780 sublines acquired the inverse resistance to PTX and CDDP—the level of resistance to PTX increased as an effect of the drug treatment and the level of resistance to CDDP decreased as a bystander effect.For both PTX and CDDP, the changes in resistance were from a few- to several dozen-fold.

The very wide spectrum of induced changes in resistance to PTX and CDDP compelled us to use the individual serial dilutions of each drug for testing the cytotoxicity in each cell line (as listed in Appendix A). The dose-response curves for the parental A2780 cell line and six sublines illustrate how a long-term treatment of A2780 cells with increasing doses of PTX results in the development of an inverse resistance to PTX and CDDP (Figure 2).

The drug concentration points were not compatible between the cell lines, therefore, we decided to perform the transformation, which consisted of the determination of drug concentrations for eight selected values of viability (i.e., 90%, 80%, 70%, 60%, 50%, 40%, 30%, and 20%) for each cell line. Two points $A\left(x_{1},y_{1}\right)$ and $B\left(x_{2},y_{2}\right)$ were chosen, where $x_{1}$ and $x_{2}$ represented the drug concentration, whereas $y_{1}$ and $y_{2}$ represented the % viability. For any chosen points A and B, the linear equation that contained the two points was determined. Next, the equation was transformed so that for any chosen value y (the % viability) it was possible to determine x (the drug concentration). The resulting formula had the following form:(1)$$x=x_{1}+\frac{\left(x_{2}-x_{1}\right)\left(y-y_{1}\right)}{\left(y_{2}-y_{1}\right)}$$

Based on this equation, the drug concentrations for the viabilities of *y* = 90%, 80%, 70%, 60%, 50%, 40%, 30%, 20% were determined. The performed transformation allowed for the next comparative analysis between the cell lines.

For further studies, six A2780 sublines were chosen. Their IC_50(120h)_ values for PTX and CDDP are presented in Table 1. The cell subline A/4PTX-63d, exhibiting about a 2.5-fold resistance to PTX in comparison with the A2780 parental cell line, opens the list. This level of resistance can be considered as clinically relevant, since studies on cell lines, that originated from ovarian cancer patients before and after chemotherapy, revealed that a 2- to 3-fold resistance was typically developed in patients after chemotherapy [10,13]. In the consecutive five sublines with tolerance towards 8, 16, 32, 64, and 128 nM PTX, the level of resistance to PTX achieved from several to several dozen-fold, therefore, it is much higher than the clinically achieved resistance. However, we expected that these highly resistant models were more helpful to understand the molecular mechanisms of resistance. The developed inverse resistance to PTX and CDDP, being in line with the most common clinical response, is the advantage of our model.

The resistance to PTX increased systematically from A/4PTX to the A/128PTX cell line, whereas the sensitivity to CDDP initially remained unchanged and then increased systematically from A/8PTX to A/64PTX and decreased in the A/128PTX cell line. However, considering the fold changes in drug resistance in the pairwise comparison of consecutive cell lines (Table 2), one can notice two clear visible breakthroughs in the PTX resistance development: The fold resistance to PTX reached higher values following treatment of the A2780 parental cell line with 4 nM PTX (2.48) and following treatment of the A/8PTX subline with 16 nM PTX (4.14). Moreover, the A/16PTX cell line exhibits the highest fold change in CDDP sensitivity (3.26). These relationships may partially explain the reason for the hard adaptation of cells to 4 and 16 nM PTX concentrations, described in Section 2.1.

#### 2.2.2. Comparison of Differences in Cell Survival

In the first approach, the dose-response curves were compared to assess whether they significantly differed by using the linear mixed models (LMM) [16]. In the analysis, it was assumed that the percentage of survived cells for a given cell line *i* in the cytotoxicity assay for PTX or CDDP was described by the equation of linear regression:(2)$$y_{i,\%\;viability}=a_{i}+b_{i}x_{i,conc.}$$
where *y_i,_*_% *viability*_ was the percentage of survival for the *i*-cell line, *x_i,conc._* was the drug concentration used in the cytotoxicity assay, and *a_i_* (intercept with vertical axis) and *b_i_* (slope) were coefficients of linear dependence. The analysis was focused on the b_i_ coefficient, which was assumed as a random effect (whereas *a_i_* was considered as fixed) and reflected the speed of changes in the survival of cells. A visualization of the equations estimated for each cell line and each drug can be found in Figure 3. As a rule of thumb, the “steeper” the approximated line, the faster the cell survival changes and the “less parallel” the lines, the more different they are. *p*-values for the dose-response relationship approximation in the cell line pairwise comparison are presented in Table 3.

In the second approach, we employed—more traditionally—the unpaired Student’s t-test for comparing the experimental means pIC_50(120h)_ values (the negative log_10_ of the IC_50_ value in the molar) to assess whether they differed. The t-test was performed on pIC_50_ values rather than commonly used IC_50_ values. Only the *p*-scale values are log normally distributed, and therefore, they allow for fulfilling the requirement of data normal distribution in a parametric t-test and performing on a small sample number, as well as the wide range of development drug resistance/sensitivity [17,18,19]. *p*-values for the mean pIC_50_ values in the cell line pairwise comparison are presented in Table 3.

The prevalent majority of both LMM and t-test results furnished the similar evaluation of statistical significance. However, one can notice that several results are inconsistent between two methods. Especially in the three cases, the differences between the two cell lines were assessed as not statistically significant in LMM, whereas, statistically significant in the t-test (Table 3). Albeit the analysis of 95% CI in two of these cases (A2780-A/4PTX and A/16PTX-A/128PTX for CDDP) reveals their trend towards significance (Appendix A). In summary, an analysis of the course of the dose-response relationship in LMM provides a more reliable evaluation of the development of drug resistance/sensitivity than the commonly employed analysis of the only one point of this curve—the IC_50_ value.

### 2.3. Gene Expression Analysis in the A2780 Parental Cell Line and Six Derived Sublines with PTX-Induced Inverse Resistance to PTX and CDDP

#### 2.3.1. Number of DEGs Is Positively Correlated with the Level of Drug Resistance

A total of 11,588 genes were detected in the A2780 cell line and six derived sublines by next generation sequencing (NGS). The pairwise comparison of the A2780 cell line and derived cell line allowed us to identify in this gene set 5810 DEGs with a false discovery rate (FDR)-adjusted *p*-value [20] (i.e., *q*-value) < 0.05 and among them 2929 DEGs with *q*-value < 0.01 were found. The number of DEGs with two thresholds: <0.01 and <0.05 was then compared in all possible 21 cell line pairs (Appendix A). Six pairwise comparisons between the A2780 parental cell line and six derived sublines are presented in Figure 4.

We demonstrated that the resistance to PTX and the sensitivity to CDDP systematically increased with the increasing of well tolerated PTX concentrations (8-16-32-64 nM PTX). In these four cell lines, the NGS analysis showed very clearly that the more the derived subline differed in drug resistance/sensitivity from the parental cell line A2780, the more differences could be observed in gene expression. Two sublines: A/32PTX and A/64PTX reached a very high resistance to PTX and the highest sensitivity to CDDP and they also had the highest number of DEGs. This trend was visible already in the first cell line (although not yet clearly)—it could be caused by the fact that the A/4PTX subline was the line which needed the longest time (63 days) to adapt to the 4 nM PTX concentration (Figure 1). However, this trend was broken in the last cell line of the series—the A/128PTX subline reached the highest resistance to PTX, but its sensitivity to CDDP declined compared to three preceding sublines. The number of DEGs in A/128PTX cells dropped dramatically in comparison to A/32PTX and A/64PTX cells (Figure 4).

To verify the trend “the higher difference in drug resistance/sensitivity the higher number of DEGs”, we performed the Spearman’s correlation analysis, which revealed the statistically significant positive correlation between the number of DEGs for 21 pairwise comparisons between seven cell lines and PTX-RI (Appendix A) and CDDP-SI (Appendix A). A positive correlation was very strong for DEGs with *q*-value < 0.01, both for PTX and CDDP and equal to 0.86 and 0.85 (*p*-value < 0.001), respectively. The comparison of one of the two cell lines: A/32PTX or A/64PTX, with any other cell line, results in the highest number of DEGs (see Appendix A). Interestingly, there is no statistically significant difference in gene expression between A/32PTX and A/64PTX. It may be suggested that the gene expression profile changed the most until the stage represented by the A/32PTX cells, then it was maintained in A/64PTX cells, and finally the effect started to reverse, which is observed as a decreased number of DEGs in the A/128PTX cells.

The observed dramatic changes in the number of DEGs and their positive correlation with the changes in PTX and CDDP resistance, revealed how globally ovarian cancer cells reconfigured their gene expression profile when they faced the anti-cancer drug treatment.

#### 2.3.2. Trends in Kinetics of Gene Expression Changes during the Development of Inverse Resistance to PTX and CDDP

To study more deeply gene expression changes in the development of inverse resistance to PTX and CDDP, we tested how many DEGs were common for particular pairwise comparisons. Altogether, we performed 30 comparisons, both for DEGs with *q*-value < 0.05 and with *q*-value < 0.01, with division into over- and under-expressed genes (Figure 5).

Looking at the bubble size in Figure 5, reflecting the number of shared DEGs, one can notice the visible difference that appeared along with the development of inverse resistance. Six cell sublines can be separated into two groups: “The early stages in inverse resistance development” (A/4PTX and A/8PTX sublines) and “the late stages in inverse resistance development” (A/16PTX, A/32PTX, A/64PTX, and A/128PTX sublines). The terms: “The early stages/sublines” and “the late stages/sublines” as used hereafter refer to these two groups. The early sublines, compared between themselves or with the late sublines, exhibited common over-expressed DEGs outnumbered by under-expressed DEGs (e.g., 244 < 335 in Figure 5A or 83 < 202 in Figure 5B). On the contrary, in all comparisons between the late sublines common over-expressed DEGs predominated over under-expressed DEGs (e.g., 1373 > 1162 in Figure 5A and 300 > 235 in Figure 5B). The same trend was observed both in the gene set with *q*-value < 0.05 and with *q*-value < 0.01. This observation suggested that more genes declined their expression at the early stages of resistance development, but at the late stages more genes elevated their expression. Presumably, the switch in the kinetics of gene expression changes occurred during the treatment of A/8PTX cells with 16 nM PTX. This presumption was formerly suggested by the observation that the A/16PTX subline presented the highest fold changes in both PTX resistance and CDDP sensitivity (Section 2.2.1).

The more closely sublines are related in the family tree (Figure 1), the more common DEGs (both up- and downregulated) tend to have, both in the gene set with *q*-value < 0.05 and *q*-value < 0.01. This trend is clearly visible for comparisons with two border sublines: A/4PTX and A/128PTX. The examples are the following numbers of shared DEGs: 131, 119, 84, 85, and 87 in the upper row of Figure 5B and: 224, 280, 381, 447, and 464 in the lowest row of Figure 5A.

To summarize, the bubble plot analysis confirmed the evolutionary character of inverse resistance development and the presence of the turning point in this process.

#### 2.3.3. Gene Expression Signature Is Able to Separate the A2780 Parental Cell Line and Six Derived Sublines

Considering the differences of the number of DEGs in the series of six sublines, which were developed from the parental A2780 cell line in the order “great-grandparent–grandparent–parent–child–grandchild–great-grandchild, etc.”, we performed a principal component analysis (PCA) to check if these cell lines with a common ancestor may be distinguishable only on the basis of gene expression signature after the PTX treatment.

PCA was performed with the use of 997 of the most significant genes with *q*-value < 0.001, selected from all 5810 identified DEGs (*q*-value < 0.05). Each one of the two biological replications for each cell line was analyzed independently (Figure 6). The first principal component (PC1) explained 62.17% of variability among all cell lines and differentiated the parental cell line and the early sublines from late sublines. PC2, reflecting 19.78% variability, diversified A2780 from early sublines and separated, though to a lesser extent, A/4PTX and A/8PTX from each other. However, A/32PTX and A/128PTX sublines were indistinguishable on the 2D PCA plot (Appendix A). The separation between those two sublines was possible with PC3 and PC4, responsible for 7.29% and 3.38% variability, respectively (Appendix A).

With the use of principal components: PC1, PC2, and PC4, we were able to reach an evident separation of seven studied cell lines (Figure 6). Looking at the data with these three different angles, we can also see that two biological replications of each cell line were practically indistinguishable. This confirms that the results of the RNA sequencing (RNA-seq) data analysis were repetitive.

#### 2.3.4. Trends in Kinetics of Gene Expression Changes Are Able to Follow the Process of Differentiation of A2780-Derived Sublines during the Development of Inverse Resistance to PTX and CDDP

Focusing on six comparisons between the A2780 cell line and each derived subline, we clustered the studied cell lines with the use of unsupervised hierarchical clustering and two sets of DEGs:997 DEGs (*q*-value < 0.001), previously selected for PCA (Figure 7).160 DEGs with *q*-value < 0.001 and absolute log_2_ of fold change (FC) > 2 (|log_2_ FC| > 2) in each pair of cell line comparison (the subset of 997 DEGs) (Figure 8). The term “log_2_ FC” as used hereafter refers to the ratio of two expression values of a gene, e.g., in a studied cell line vs. control cell line. For example, |log_2_ FC| values > 2.0 are equivalent with more than the 4-fold expression change.

In both cases, hierarchical clustering separated all sublines into two classes: The first containing the early sublines and the second containing the late sublines. Additionally, the subgroup of three sublines at the late stages: A/32PTX, A/64PTX, and A/128PTX were grouped in one subcluster. This suggests that those three sublines, representing the advanced stages in inverse resistance development, present generally a similar pattern of gene expression. Hence, the intermediate subline A/16PTX played a role of the bond between them and the early sublines. It is worth noting that the separation between the A/16PTX subline and A/32PTX, A/64PTX, and A/128PTX sublines was also clearly visible in PCA with PC1 and PC2 components (Appendix A). Therefore, the terms: “The intermediate subline/stage” and “the advanced subline/stage” as used hereafter refers to the A/16PTX subline/stage and to A/32PTX, A/64PTX, and A/128PTX sublines/stages, respectively.

The heat maps revealed five expression patterns of DEGs:DEGs upregulated in six comparisons (consistently upregulated genes).DEGs downregulated in six comparisons (consistently downregulated genes).DEGs downregulated in the beginning and then upregulated (genes with a rising trend).DEGs upregulated in the beginning and then downregulated (genes with a descending trend).DEGs with variable expression, which cannot be classified to any of the four classes above (genes with variable expression).

In the set of 997 DEGs, the number of consistently upregulated and consistently downregulated genes were similar (190 and 200 DEGs, respectively). The total number of genes with rising and descending trends (469 DEGs) was larger than the total number of genes with consistently changed expression (390 DEGs). The variable expression was exhibited by 13.84% DEGs (Figure 7).

In the subset of 160 DEGs, the proportion was different: Genes with consistently changed expression (148 DEGs) made up the prevalent majority over genes with regularly and irregularly changing expression (12 DEGs). The variable expression was exhibited only by one DEG (0.62%) (Figure 8).

Noteworthy, in both sets of DEGs (997 and 160), the descending trend in gene expression is dominated by a rising trend (174 vs. 295 and 1 vs. 10, respectively), which confirms the tendency revealed by the bubble plot analysis performed on two bigger sets of DEGs with higher *q*-values (Section 2.3.2): At the early stages of resistance acquiring more genes declined their expression, whereas at the late stages more genes elevated their expression and the development of the A/16PTX subline is a switch point in the kinetics of gene expression.

In summary, the imposing of a more stringent criterion (|log_2_ FC| > 2) on DEGs with *q*-value < 0.001 resulted in the selection of a subset of DEGs with coherent changes in expression during the development of inverse resistance. Hence, we used 160 DEG subsets as a starting point for further analysis. The terms: “DEGs”, “downregulated genes”, and “upregulated genes” as used hereafter refer to the results of gene expression comparison between the A2780-derived cell subline and A2780 cell line.

### 2.4. Scaffold Functional Interaction Networks of Proteins Encoded by DEGs, Identified in Each One of Six A2780-Derived Sublines Compared to the Parental Cell Line

We built a PPI network of proteins encoded by 160 DEGs, previously visualized as a heat map (Figure 8) and forming a scaffold network during a gradually developing inverse resistance to PTX and CDDP. We identified two functional networks of 53 proteins and 11 proteins and several small groups of 2–3 proteins (only two of them are presented) (Figure 9). It is worth noting that the 53 protein network was formed in a prevalent majority by proteins encoded by downregulated genes. Seven proteins in this network may be recognized as hubs (with 7–11 interactions with partners): SRY(sex-determining regionY)-box 9 (SOX9); RUNX family transcription factor 2 (RUNX2); collagen type I alpha 2 chain (COL1A2); thy-1 cell surface antigen (THY1); zinc family member 1 (ZIC1); Periostin (POSTN), and neurexin I-alpha (NRXN1). All hubs are encoded by downregulated genes.

Proteins in these networks are enriched for biological processes, which are connected with the nervous system development, ossification, development, and metabolic processes of cartilage and cell adhesion, as well as the tumor necrosis factor (TNF), ephrin, and nuclear factor kappa B subunit 1 (NFKB1) signaling. The detailed lists of enriched GO biological processes and involved genes are provided in Appendix A. It is worth mentioning that the prevalent majority of proteins in these networks are enriched for two processes connected with: (1) Nervous system development and (2) chondro- and osteogenesis.

## 3. Discussion

This paper presents the comprehensive transcriptome analysis of a series of six human ovarian cancer isogenic cell lines with a gradually changing inverse resistance to PTX and CDDP. The series was established by the chronical exposition of parental A2780 cells to stepwise increased concentrations of PTX. Within the increasing resistance to PTX, cells collaterally acquired sensitivity to CDDP.

As the process of resistance development is governed by a natural selection, it is difficult to predict and control the pattern of in vitro developed resistance. In the process of resistance induction in different ovarian cancer cell lines: 2008 [21,22], KF28 [23], UL-3A [24], OV90 [25], W1 [26], and A2780 [27,28,29] various PTX administration schedules were employed. Chronical [21,23,25,26,28,29] or pulse [22,24,27] exposure in time to constant [24] or stepwise increasing [21,22,23,26,27,28,29] drug doses resulted in acquiring of cross- [24,25,26,27,28,29] or inverse- [21,22,23] resistance to PTX and CDDP. There was no rule that the specific regimen led to a specific pattern of resistance—visibly it was the issue of randomness of natural selection. It is worth noting that Li et al. [27], Wang et al. [28], and Januchowski et al. [29] using the incrementally increasing PTX doses in A2780 cells generated the sublines resistant both to PTX and collaterally to CDDP (cross resistant). Our team, employing a similar regimen of the PTX treatment, developed A2780 sublines resistant to PTX and collaterally sensitive to CDDP.

The cell culture model, established by our team, is valuable both from the clinical and biological point of view. First, it is compatible with the trend to acquire inverse resistance between taxanes and platinum derivatives, observed in clinical practice [10]. Moreover, the cellular and molecular mechanisms of this response have not been unraveled yet. Our cell line series provides the unique possibility of keeping track of trends in gene expression involved in this phenomenon and may be applied in further studies of drug resistance evolution in vitro.

The great number of identified genes with a significantly altered expression in six cell sublines compared to the parental cell line (5810 DEGs with *q*-value < 0.05 among a total of 11,588 detected genes) revealed how deeply cancer cells needed to rearrange their gene expression to acquire drug resistance. The positive correlation of the total number of DEGs with the changes in PTX and CDDP resistance (Appendix A) suggests the evolutionary character of this process.

Our analyses argued for the scenario that individual genes and groups of genes were enrolled (i.e., differentially expressed) or excluded (i.e., their expression came back to constant) at the consecutive stages of resistance acquiring. Other genes might constitute a scaffold of the process, i.e., they were consistently over- or under-expressed at each stage of resistance development or their expression changed in a consistent direction. We hypothesized that these different patterns of gene expression reflect their role in resistance acquisition. Taking advantage of the in silico analysis of functional interactions of proteins encoded by those genes, we identified two main hubs in the scaffold networks: SOX9 and RUNX2. These proteins were engaged in two most enriched biological processes, connected with (1) nervous system development, and (2) connective tissue, especially in bone and cartilage development and metabolism.

SOX9 and RUNX2 are evolutionary old transcription factors, so-called master transcriptional regulators, due to their governing role in the decision of multipotent mesenchymal stem cells (MSCs) fate in embryogenesis and adult homeostasis. The mutual relationship between SOX9 and RUNX2 is especially intimate in the skeletogenesis. The SOX9 primary commits MSCs to cartilage forming chondroblast lineage, whereas RUNX2 to bone-forming osteogenic lineage. In the endochondral ossification, where bone formation is mediated through chondrocytes, SOX9 and RUNX2 act in a cooperative or antagonistic way, tightly regulating consecutive steps in this complex process [30,31,32,33].

The prominent role of SOX9 and RUNX2 in the normal development of ectodermal and mesodermal originated tissues makes them particularly prone to being hijacked by cancer cells and exploited during neoplastic initiation, development, and progression. Indeed, both SOX9 and RUNX2 can act as oncogenes or tumor suppressors in a context-dependent manner—their expression is characteristic for cancer stem-like cells (CSCs) and they are involved in metastasis and chemoresistance [34,35]. In particular, the over-expression of RUNX2 promotes prostate and breast cancer metastasis [36,37].

Recently, several reports (prevalently microRNA studies) have indicated that the elevated expression of RUNX2 [38,39,40,41] or SOX9 [42,43,44,45,46] conferred resistance to taxanes (prevalently docetaxel) [38,40,41,42] and CDDP [39,43,44,45,46] in prostate [38,41,42], gastric [43,45], lung [39,46], breast [40], and ovarian [44] cancer cells and tissues. In our experiments, the expression of both genes was consistently downregulated at all stages of inverse resistance development. We might say that as for CDDP, our results are congruent with published findings and as for PTX, they are disparate. However, this interpretation would be too simplified, since in the studies cited above, the status of only one of those genes was investigated.

Several other proteins in the scaffold networks, which are engaged in osteo- and chondrogenesis, attract attention. There are six extracellular matrix (ECM) proteins. COL1A2 is the main structural protein of the bone ECM [47]. POSTN is an osteoblast specific factor, produced by fibroblasts and secreted to ECM during bone and teeth development, notably produced also by cancer cells and associated with disease progression and drug resistance [48,49]. Matrix Gla protein (MGP) is secreted by chondrocytes [50] and inhibits cartilage calcification [51], as well as reinforced angiogenesis that can favor tumor progression [52]. The C-type lectin domain containing 11A (CLEC11A, alias osteolectin) stimulates the differentiation of MSCs into mature osteoblasts [53,54]. SPARC, cwcv, and kazal-like domains proteoglycan 2 and 3 (SPOCK2 and SPOCK3, alias testican-2 and testican-3), two members of secreted protein acidic and cysteine rich (SPARC) family, are matricellular proteoglycans [55]. They show opposite functions: SPOCK3 inhibits the activity of membrane-type metalloproteinases (MT-MMPs)—zinc endopeptidases that cleave collagens, proteoglycans, and many other components of ECM, mediating tumor invasion and metastasis [56]. SPOCK2 counters this SPOCK3 action and thereby allows for MT-MMP mediated cell migration. Therefore, in cancer disease, SPOCK3 reveals an anti-tumor invasion activity, whereas SPOCK2 reveals a pro-tumor invasion activity [57,58].

The next bone-related protein is vascular endothelial growth factor C (VEGFC) belonging to crucial factors in embryonal and tumor angiogenesis processes, as well as involved in vascularization during bone formation [59]. Moreover, two transcription factors: Myocyte enhancer factor 2C (MEF2C) and distal-less homeobox 5 (DLX5) enhance the expression of *RUNX2* [60]. Another protein, chordin (CHRD) acts as an antagonist of bone morphogenetic protein (BMP) signaling pathway—one of the main developmental pathways, strongly involved in bone and cartilage development [61]. Finally, osteocrin (OSTN) is secreted by osteoblasts and inhibits their differentiation [62]. The insulin-like growth factor binding protein 3 (*IGFBP3*), independently of its primary role of insulin growth factor carrier, is involved in chondrogenesis. IGFBP3 inhibits the growth of chondroprogenitors by inducing MSC apoptosis and antagonizes transforming growth factor beta 1 (TGFB1) chondro-inductive effects [63]. In our study, among 14 identified DEGs engaged in ossification and chondrogenesis, only five genes: *SPOCK2*, *SPOCK3*, *CHRD*, *OSTN*, and *IGFBP3* were found consistently upregulated and the 6th one—*CLEC11A* showed the rising trend in its expression.

Four other genes in our networks: *THY1*, *ZIC1*, *EPHA5*, and pyroglutamylated RF amide peptide receptor (*QRFPR)* encode proteins of versatile activities, including bone formation and function, although their roles in these processes have not been extensively explored. The upregulated *QRFPR* gene encoded GPCR for neuropeptide QRFP is involved in energy homeostasis and hormone secretion and promotes early stages of endochondral bone formation [64,65]. Versatile activities of three other genes are presented below.

The over-representation of bone-related proteins in our scaffold networks and the fact that in a prevalent majority they are encoded by under-expressed genes, are striking. It brings to mind the possible association with the osteomimicry phenomenon. Osteomimicry is the ability of tumor cells to exhibit a gene expression profile that resembles the profile of resident bone cells, i.e., osteoblasts and osteoclasts, for better survival in the foreign environment and intruding in the bone physiology that facilitates metastazing. This phenomenon was first described in prostate and breast cancers, preferentially metastasize to bone, where RUNX2 is a master regulator during the transformation of cancer cells into osteomimetic cells [66]. Our results suggest that the PTX treatment and acquiring of inverse resistance to PTX and CDDP may turn off (e.g., via negatively regulated RUNX2) the osteomimic phenotype in ovarian cancers cells, at least in vitro.

Three rest hubs in the 53 protein network: THY1, ZIC1, and NRXN1, encoded by consistently downregulated genes, are connected with the nervous system. NRXN1 is a cell adhesion protein, involved in synaptic signal transmission [67]. Activities of THY1 and ZIC1 are much more broad and complex.

The upregulated *THY1* gene encodes a small cell surface glycoprotein, acting as a ligand both *in trans* and *in cis* configuration; regulating differentiation in cells of ectodermal, endodermal, and mesodermal origin—in summary this small protein controls the extraordinary diversity of biological processes [68]. THY1 plays a key role in the fate decision of MSCs—it promotes osteogenesis (e.g., by supporting the *RUNX2* expression) and inhibits adipogenesis [69,70]. THY1 also regulates cancer cell proliferation, tumor metastasis, and angiogenesis, playing a dual role as a tumor promoter or suppressor. Notably, in ovarian cancer it functions as a tumor suppressor and its downregulation is associated with poor prognosis [71].

Downregulated *ZIC1* and *ZIC4*, as well as *ZIC2* which showed the rising trend in expression, encode zinc-finger transcription factors, play a crucial role in neural, muscle, and bone development [72]. They are also deregulated in many cancers [73]; particularly ZIC1 takes part in the bone’s response to mechanical stress [74] and its expression is increased in a malignant form of epithelial ovarian cancer and strongly associated with clinical outcome [75].

Four proteins in the scaffold networks belong to the TNF ligand and receptor superfamilies: TNF superfamily member 10 (TNFSF10), TNF receptor superfamily member 10d (TNFRSF10D), CD 40 molecule (CD40), and CD70 molecule (CD70). TNFSF10 is a tumor necrosis factor-related apoptosis-inducing ligand (TRAIL) and TNFRSF10D is one of its receptors. In contrast to TNF, which induces apoptosis both in cancer and healthy cells and thereby its administration in cancer patients causes unacceptable systemic toxicity, TRAIL selectively affects tumor cell survival without harming healthy cells [76]. Therefore, in the last two decades, TRAIL and its receptors have been explored as an attractive drug target.

TNFSF10 binds to five cognate receptors: TNFRSF10A, TNFRSF10B, TNFRSF10C, TNFRSF10D, and TNFRSF11B. Upon binding two receptors with a complete cytoplasmic death domain (TNFRSF10A or TNFRSF10B), TNFSF10 can trigger a dual action: Canonical caspase-dependent apoptosis (in cancer cells) or non-canonical cell survival/proliferation pathway (both in cancer and healthy cells), e.g., via NFKB1 signaling, which also stimulates migration and invasion of cancer cells. The second pathway makes some tumor cells TRAIL-resistant, which is counterproductive for oncological therapy, but can be overcome by a combined treatment with chemotherapy. The three next receptors (including TNFRSF10D) lack the functional death domain and act as decoy receptors, sequestering TNFSF10 and preventing apoptosis. The presence and abundance of decoy receptors impact the cell response to TRAIL [77]. Therefore, both *TNFSF10*, consistently upregulated and *TNFRSF10D*, consistently downregulated in the scaffold network, play a dual role of tumor suppressors and promoters.

In several studies, TRAIL was found as a factor increasing the sensitivity to cytotoxic drugs in different cancer types. Especially, Vignati et al. demonstrated that although TRAIL did not significantly reduce the growth of A2780 cells, in combination with CDDP or PTX it did boost the apoptosis of these cells, albeit this synergism did not seem to be connected with the TRAIL receptor status [78]. Our results support the positive correlation between *TNFSF10* expression and sensitizing A2780 cells to CDDP, but on the other hand, reveal the negative correlation in sensitizing A2780 cells to PTX. The declined expression of *TNFRSF10D* suggests that the TNFSF10 activity is not suppressed by a decoy receptor.

The CD40 molecule (CD40) is a receptor (its native ligand is CD40 ligand (CD40L)), whereas CD70 is a membrane-bound ligand (its native receptor is the CD27 molecule (CD27)). CD40 and CD70 are detected on a wide range of cancer cells (including ovarian cancer cells) and cells from their microenvironment [79,80]. In breast carcinoma, CD40L is expressed on non-cancer cells from a tumor environment and protects CD40+ cancer cells through a caspase-independent pathway against apoptosis caused by cytotoxic agents (e.g., CDDP and PTX) and as a result, induces a multidrug resistance [81]. Here, *CD40* is downregulated in all A2780 sublines with elevated resistance to PTX and declined resistance to CDDP, therefore, our findings seem to be in line with published data as for CDDP, but not as for PTX.

The CD70-CD27 axis inhibits an anti-tumor response of tumor-infiltrating lymphocytes. CD70+ cancer cells communicate with CD27+ tumor infiltrating lymphocytes and induce their apoptosis. Therefore, the constitutive CD70 expression on tumor cells lets them evade the elimination by the host immune system [80]. In our research, *CD70* showed the rising trend in expression, which may suggest that in the later stages of resistance development, A2780 cell sublines improved their ability to avoid destruction by the immune system.

Three ephrin receptors: EPH receptor A3 (EPHA3), EPH receptor A5 (EPHA5), and EPH receptor A7 (EPHA7) were found in the scaffold network. Both EPHs (tyrosine kinases, conventional receptors) and ephrins (conventional ligands) are membrane tethered proteins, involved in short-distance cell-cell signaling. They act simultaneously as receptors and ligands in two neighboring cells and signaling events mediated by them are extraordinary plastic. They can send “forward signaling” in the EPH-bound cell and “reverse signaling” in the ephrin-bound cell; they can work both *in trans* and *in cis* configuration. Moreover, both EPH and ephrins themselves (without ligand or receptor binding) can transmit a signal into the cell. Ephrins and their receptors regulate many developmental processes: Both in the nervous system and in the differentiation of MSCs and bone formation; they are also engaged in tumorigenesis by controlling tissue architecture and cell migration [82]. In particular, EPHA5 is a candidate for the negative regulator of MCS differentiation into osteocytes [83] and EPHA3 plays a proto-oncogenic role in multiple tumors [84]. Here, *EPHA5* was found consistently upregulated (which supports our former suggestion about turning off the osteomimic phenotype during the development of inverse resistance to PTX and CDDP), whereas *EPHA3* and *EPHA7* were consistently downregulated.

Another upregulated gene, polo-like kinase 2 (*PLK2*), although the protein encoded by it was identified as a modest player in the scaffold network (having only two partners), is a very interesting gene, since it was identified as the most significant DEG (with the lowest *q*-value) at each stage of inverse resistance development in our cell subline series. PLK2 is an evolutionary conserved centrosomal protein, which regulates centriole duplication, so it is responsible for an appropriate spindle function during cell division [85].

Burns et al. have shown that *PLK2* is a transcriptional target for the tumor protein p53 (TP53) and its mediator in conferring resistance to PTX. After exposition of the tumor cells to PTX, the wild type TP53 activated transcription of PLK2, which in response to spindle damage induced cell cycle arrest and prevented cell death, led to acquiring resistance to this anti-microtubule agent. The mutation or loss of TP53 had a similar effect such as silencing of PLK2—both promoted cell death after the PTX treatment and sensitized cells to this drug. The authors proposed the disruption of the PLK2 activity as an interesting idea for a new anti-cancer drug [86]. These results are congruent with our findings as for the PTX resistance displayed by the A2780 cell sublines correlated with serially upregulated *PLK2* (A2780 cells bear wild type *TP53* [87]).

This study also revealed a two-faced nature of PLK2: In a healthy cell, it plays a role of a tumor suppressor—by its involvement in the spindle checkpoint it ensures a genetic stability and inhibits neoplastic transformation. In a cancer cell, this activity changes its face—the antiproliferative impact of PLK2 during mitosis leads to drug resistance and its role reverses to pro-oncogenic.

On the contrary, later studies by Syed et al. on PLK2 have shown that in A2780 cell sublines with acquired cross-resistance to PTX (RI: 354–615) and CDDP (RI: 4.7–9.3), PLK2 was downregulated on mRNA and protein levels, which led to the loss of cell cycle control and abrogation of apoptosis. Conversely, the ectopic expression of *PLK2* restored cell cycle arrest, reactivated apoptosis triggered by PTX, and re-sensitized these cells to PTX. Therefore, the authors proposed the PLK2 status as a clinically important marker of chemosensitivity in the ovarian cancer treatment [88]. These results are not in accordance with our findings. It may be partially explained by a different profile of resistance in our cell line series—they are much less resistant to PTX (RI: 2.5–59) and are sensitive to CDDP. Moreover, it seems that the involvement of PLK2 in drug resistance development may be contextual and rather ambiguous.

PLK2 is also active in postmitotic cells such as neurons. Neuronal PLK2 contributes to the maintenance of global synaptic homeostasis [89]. On the other hand, anti-cancer agents such as PTX and CDDP disrupt the nervous system functionality by chemotherapy-induced peripheral neuropathy, axon degeneration due to microtubules stabilization by PTX in sensory neurons (a frequent side effect of treatment) [90,91]. EPHA5, another protein in our scaffold network encoded by the upregulated gene and engaged in the nervous system development, was recently identified as associated with PTX-induced neuropathy in cancer patients [92,93]. The compelling question concerning the possible interaction between PLK2, EPHA5, and PTX effects on neurons is still a matter of speculation.

In summary, our cellular model and transcriptome-focused approach suggests the evolutionary character of drug resistance development and indicates some unexpected factors and signaling pathways, which may be involved in this process. The scaffold DEGs encode proteins which present specific features: (1) Some of them are tumor suppressors (such as *PLK2* and *SPOCK3*) or tumor promoters (such as *SPOCK2* and *EPHA3*), but the majority plays a dual role of promoters and suppressors in a context-dependent manner (*SOX9*, *RUNX2*, *THY1*, *TNFSF10*, and *TNFRSF10D*); (2) some of them control unusually diverse biological processes and mediate signals in an extraordinary plastic way (such as *THY1* and *EPHAs*); (3) some of them are cell cycle regulators (such as *PLK2*); (4) some of them can trigger a dual action: Apoptosis or survival in a context-dependent manner (such as *TNFSF10*, *CD40*, and *CD70*); (5) strikingly many DEGs are linked to osteo- and chondrogenesis (such as downregulated *SOX9*, *RUNX2*, *COL1A2*, *POSTN*, *MGP*, *VEGFC*, *MEF2C*, *DLX5*, *THY1*, *ZIC1* and upregulated *SPOCK2*, *SPOCK3*, *CHRD*, *OSTN*, *IGFBP3*, *EPHA5*, *QRFPR*) and a vast majority of them are under-expressed. Moreover, CHRD, OSTN, IGFBP3, and EPHA5 proteins, encoded by over-expressed genes, function as negative regulators of osteo- and chondrogenesis. This picture suggests the possibility of massive turning off of osteo- and chondrogenesis associated genes during resistance development.

The complexity and the broad spectrum of scaffold DEG activities make the interpretation of our findings very difficult and ambiguous. At this stage of the research, one should not interpret the correlation of gene expression with resistance as a simple cause-result relationship. Obviously, we cannot exclude that during the development of the series of six cell lines over a long time period (17 months), other molecular and cellular processes were involved in the selection of cells surviving increasing PTX doses. In a non-homogenous population of the A2780 cell line, drug-sensitive cells could acquire resistance, but also the selection of an initially resistant subpopulation could take place. Some events, e.g., acquired mutations or changes in gene expression, might promote the resistance development, whereas others may be only accompanying. The real contribution of activated genes and pathways in PTX resistance and CDDP sensitivity must be validated by functional studies.

Nevertheless, the results of an extensive comparative analysis of transcriptomes, presented in this paper, point out the directions for further research. To our knowledge, this study for the first time raises the issue of the plausible phenomenon of abrogation of the osteomimic phenotype of ovarian cancer cells during development of inverse resistance between PTX and CDDP. Until this time, osteomimicry has been associated rather with the tendency of breast and prostate cancer cells to metastase to bones, but not with drug resistance. However, to get a more complete overview and explore this phenomenon more deeply, further studies are necessary.

## 4. Materials and Methods

### 4.1. Cell Line and Reagents

The human ovarian cancer cell line A2780 of endometroid carcinoma histotype [87] was purchased from the European Collection of Authenticated Cell Cultures (ECACC, Porton Down, Salisbury, UK, cat. no. 93112519). Cells were grown in a monolayer and conventionally were maintained in the RPMI 1640 medium with phenol red (unless otherwise indicated), supplemented with a 5% heat-inactivated fetal bovine serum (FBS) (Life Technologies—Thermo Fisher Scientific, Waltham, MA, USA), without antibiotics and antimycotics, at 37 °C and 5% CO_2_. Cell cultures were periodically monitored for mycoplasma contamination using the PCR mycoplasma test kit for the qualitative conventional PCR (PromoKine, Heildelberg, Germany) according to the manufacturers’ instructions. All cultures used in the experiments were mycoplasma free.

AlamarBlue (aBlue) was purchased from Life Technologies—Thermo Fisher Scientific. CDDP, PTX, phosphate-buffered saline (PBS), and other chemicals were purchased from Sigma-Aldrich (St. Louis, MO, USA). The 0.2 mM stock solution of PTX was prepared in DMSO and the 7.5 mM stock solution of CDDP was prepared in PBS; stock solutions were aliquot and stored in −20 °C. The solubility of CDDP was augmented by warming the stock solution to 40 °C [94].

### 4.2. Generation of the Drug Resistant Series of A2780 Sublines

Cell sublines derived from the A2780 parental cell line were developed by growing A2780 cells in step-wise increases in PTX doses following general instructions, according to Coley [95]. The treatment was commenced at around 5% of IC_50 (120 h)_ for A2780 cells.

A2780 cells were seeded at 1.6 × 10^3^/cm^2^ viable cell density to a tissue culture flask and allowed to adhere for 24 h. The next day, when the culture was about 20% confluent, PTX was added to 0.25 nM of the final concentration in the medium and cells were grown until confluency became about 90% (meanwhile the medium with PTX was changed every 2–3 days). Then, the next cycle of treatment of the double PTX dose (0.5 nM) was initiated. This procedure was continued, generally followed by the pattern of doubling the PTX concentration after cells reached the stable tolerance to the former dose.

When the cells appeared not to have tolerated the consecutive PTX dose, they were allowed to grow in a drug-free medium for recovery, i.e., until growth appeared healthy and confluency reached about 90%. Then, the cells were passaged to a fresh culture flask at 1.6 × 10^3^/cm^2^ viable cell density and retreated with the formerly non-tolerated PTX dose until the stable tolerance was reached. Only then, the PTX dose was doubled. In some cases, the cells were able to bear the subsequent, double dose of PTX several days earlier before achieving the stable tolerance to the former dose—in these cases, the cells were divided into two cultures: In one flask, the treatment with the former dose was continued, whereas in the second flask, the cells were treated with the higher dose. This schedule allowed for time and cost saving.

### 4.3. Cytotoxicity Assay

The level of resistance to PTX and CDDP in parental A2780 cells and their drug resistance derivatives was determined with the aBlue assay [96]. The cytotoxicity test was set according to Plumb [97].

A2780-derived sublines with PTX-induced resistance were maintained for 3–4 days before plating in the absence of PTX. Cells were plated into flat, bottomed 96-well plates at a density of 3 × 10^3^ viable cells in 200 µL of cell suspension in a culture medium per well (columns 2–11). The edge columns (1 and 12) were blanks (BLK), i.e., 200 µL of the culture medium without cells/well. Cells were allowed to adhere and grow for 48 h. Then, the culture medium was withdrawn and eight appropriate serial dilutions of PTX or CDDP were added. Since the level of resistance to both drugs in the developed cell sublines changed from a few to several dozen-fold in comparison with the A2780 cells, it was necessary to use the individual drug concentration range for each cell line (provided in Appendix A) to cover the area of interest and precisely determine the cytotoxicity of each drug.

Drug serial dilutions in the medium were added at 200 µL/well (columns 3–10). Each concentration of the drug was added to four wells. Columns 2 and 11 were the positive growth control wells (drug-free controls): The medium was supplemented only with appropriate diluents used to solubilize the drug (i.e., with PBS for the CDDP assay and with DMSO for the PTX assay) was plated. The concentrations of PBS and DMSO were equal to the concentrations of dilution agents in wells with the highest concentrations of the appropriate drug (wells in column 10). In addition, DMSO has never exceeded a 1% concentration—the upper level of DMSO in the culture medium tolerated by the cells. Plates were incubated with drugs for 5 days (120 h) due to the diverse rate of growth among A2780 sublines—such a long period of exposure allowed each line to be treated by the drug during at least one cell cycle.

Then, the medium was withdrawn, wells were washed with PBS to remove the rest of the drugs and diluents, and then cells were fed with 200 µL of the fresh, drug-free medium without phenol red. Plates were incubated for 3 days (72 h) to allow for discrimination between viable and proliferating cells and still viable but not able to proliferate cells. This trick allowed alleviating the drawback of the aBlue test—its disability to distinguish between a cytotoxic and cytostatic drug effect.

Then, the aBlue reagent was added to each well to a 10% final concentration and after 24 h of incubation absorbance was measured at 570 and 600 nm using an EL_x_800 Universal Microplate Reader (Bio-Tek Instruments, Winooski, VT, USA).

The cytotoxicity assay for each cell line was performed on three separate plates in three biological replications. The calculation of the percent difference in the reduction of the aBlue reagent between drug-treated and control cells (equated with the percent of viability) was performed according to the equation provided by Lancaster and Fields [96].

All the analysis steps connected with the comparison of differences in cell survival: LMM and the t-test were performed with the R software [98] and Bioconductor platform. In the case of LMM and in post hoc tests, we used “lme4” [99] and “arm” [100] R packages. Graphics connected to this part of the research were made with the R package “ggplot2” [101] and Microsoft Excel (ver. 2007 and ver. 2011 for Mac, Microsoft, Redmond, DC, USA).

### 4.4. RNA-Seq of Transcripts by the NGS Technology

#### 4.4.1. Total RNA Isolation and Assessment of Its Integrity and Concentration

Total RNA was prepared from the control parental cell line A2780 and six sublines in two independent replications from each cell line. Total RNA was isolated from 5 × 10^6^ cells using the RNeasy Mini Kit (Qiagen, Hilden, Germany), based on lysing cells in the RLT buffer containing guanidine isothiocyanate, purification of the product on a spin column with a silica-gel membrane, and the treatment by DNaseI to remove genomic DNA contamination. The RNA integrity number (RIN) was analyzed using the Bioanalyzer 2100 and RNA 6000 Nano Kit (Agilent Technologies, Santa Clara, USA). The RNA concentration was measured using the fluorometer Qubit and Qubit RNA BR Assay Kit (Life Technologies—Thermo Fisher Scientific). All procedures were performed according to the manufacturers’ instructions. A total of fourteen RNA samples met the quality criteria: Their RIN range was 9.80–10, the concentration range was 0.5–1.5 µg/µL, and total RNA yield was 15–50 µg per sample, therefore, they were suitable for RNA-seq.

#### 4.4.2. NGS Libraries Preparation and Validation

Fourteen NGS libraries were prepared following the protocol of the TrueSeq RNA Sample Preparation Kit v2 (Illumina, San Diego, CA). Libraries were qualified using the Bioanalyzer 2100 and DNA 1000 Kit (Agilent Technologies, Santa Clara, USA), according to the manufacturers’ instructions. In all libraries, the final product was an anticipated band at approximately 270 bp. The total concentration of the libraries was measured using the fluorometer Qubit and Qubit dsDNA BR Assay Kit (Life Technologies—Thermo Fisher Scientific), according to the manufacturers’ instructions. Libraries were additionally quantified with qPCR according to the Illumina recommended protocol, using Rotor-Gene Q (Qiagen) and KAPA SYBR FAST qPCR Master Mix Universal (Kapa Biosystems, Wilmington, MA, USA).

#### 4.4.3. Transcriptome Sequencing and Data Analysis

Validated libraries were diluted to a 10 nM concentration, then pooled in batches of two differentially indexed samples in equal amounts, denatured, and diluted to a 10 pM concentration. The cluster generation was performed in cBot (Illumina), where the libraries were bound with a flow cell (two libraries per lane).

Sequencing by synthesis (SBS) was performed with the Genome Analyzer IIx (Illumina). Technical replicates of the libraries were clustered and sequenced with the use of different flow cells, but with the same 75-cycle multiplexed single-read sequencing recipe. Finally, we obtained approximately 25 million 75 nt-long reads per library.

Raw reads were next filtered according to the quality with the use of the FASTX-Toolkit [102]. Reads consisted of less than 80% nucleotides and at least 28 qualities were discarded. On average, 15% of the reads were filtered out per library. The remaining reads were mapped to the human reference genome version hg19 with TopHat2 [103]. On average, 96% of the reads that remained after filtering were mapped to the genome. After aligning the reads, the number of reads (counts) mapped to the known genes were obtained with the use of the featureCounts function implemented in the R package “Rsubread” [104]. All genes with very low expression values across the examined samples were discarded—only genes with more than five normalized reads mapped to the gene for at least two samples were included for further analysis.

The differential expression analysis was performed using linear models implemented in the “limma” package for R [105]. The counts for every gene were normalized with the voom function, which calculates the log_2_ value of counts, estimates their mean and variance relationship, and generates a precision weight for each observation. Next, the linear model was fitted to the data set including expression values for each gene (using the lmfit function) and empirical Bayes statistics (eBayes function) was used to improve the power of a test used for differential expression detection in sets with a high number of genes and a small number of replicates. The transcriptome of each cell line was next compared to the transcriptomes of the other six lines in a series of pairwise comparisons. The genes with *q*-values [20] below 0.05 were selected for further analysis.

The correlation analysis of the number of DEGs and PTX-RI or CDDP-SI were performed with the R software. The scatterplots were generated with the R package “ggplot2”. The same package was used for generating heat maps with hierarchical clustering, in particular, the heatmap.2 function was used [101]. Bubble plots presenting common DEGs between different cell lines comparisons were generated with the “corrplot” R library [106]. For PCA, we used the “stats” package and PCA plots were printed with the use of the plot function (2D plot) or the “scatterplot3d” package (3D plot) [107].

The mRNA-seq dataset, discussed in this paper, have been deposited in the National Center for Biotechnology Information Gene Expression Omnibus (GEO) [108] at GEO Series accession number GSE159791.

### 4.5. Functional-Interaction Network Analysis

The data analysis of RNA-seq provided the official symbol for each DEG, according to the HUGO Gene Nomenclature Committee (HGNC). Gene symbols were translated into UniProt accession numbers using the manually annotated SwissProt database—a section of UniProt Knowledgebase (UniProtKB) [109]. PPI networks were retrieved from the STRING (Search Tool for the Retrieval of Interacting Genes) database [110,111] using the list of protein accession numbers as a query and then analyzed using the Cytoscape software [112,113].

Nodes of the network represent proteins encoded by DEGs, whereas the edges illustrate the interactions among them and the intensity of edges reflecting the strength of the interaction score. The node interiors of downregulated genes are blue, whereas the node interiors of upregulated genes are red. The node size allows for a quick identification of hubs in the network—the bigger the node, the more interactions the protein has with other proteins.

Functional enrichments were retrieved from The Gene Ontology (GO) Resource [114,115,116,117] and visualized as color split charts around the nodes. For further analysis, we have chosen the enriched GO terms which had a *q*-value lower than 0.05.

The protein families, which are potential drug targets, were mapped in networks from the PHAROS database [118]. Four target families are visualized as a label color: Kinases are depicted with blue, GPCRs with red, and ion channels with green.

## 5. Conclusions

This study presents the results of a comprehensive transcriptomic analysis of human ovarian cancer cell lines with gradually developed resistance to PTX and collateral sensitivity to CDDP (inverse resistance). The series of six sublines, generated from the A2780 parental cell line, provides a unique opportunity to keep track of trends in gene expression and study this phenomenon as a process of evolution. Developmentally regulated fluctuations in gene expression, as well as the consistently changed expression of genes during the process may suggest their functional role in resistance acquisition.

The in silico analysis of functional interactions of proteins encoded by those genes indicated that, in addition to the known cancer- and drug resistance-related factors, proteins linked to the nervous system development, as well as osteo- and chondrogenesis were over-represented. We could observe the mass under-expression of genes, encoding positive regulators of bone and cartilage development and over-expression of genes, encoding negative regulators of these developmental processes. Therefore, we hypothesized that the abrogation of osteomimic phenotype in ovarian cancer cells might occur during the development of inverse resistance between PTX and CDDP. To date, osteomimicry has been rather associated with the tendency of breast and prostate cancer cells to metastase to bones, but not with drug resistance. Then, our approach allowed for the identification of new candidate genes and pathways potentially linked to drug resistance, different from the widely known “classical” factors, and hints for further functional studies.

## 6. Patents

Patent no.: PAT.233178. Date of issue: 17.06.2019. Title: “Human ovarian cancer cell culture model for paclitaxel-induced inverse resistance to paclitaxel and cisplatin and use thereof”.

## Figures and Tables

**Figure 1 ijms-21-09218-f001:**
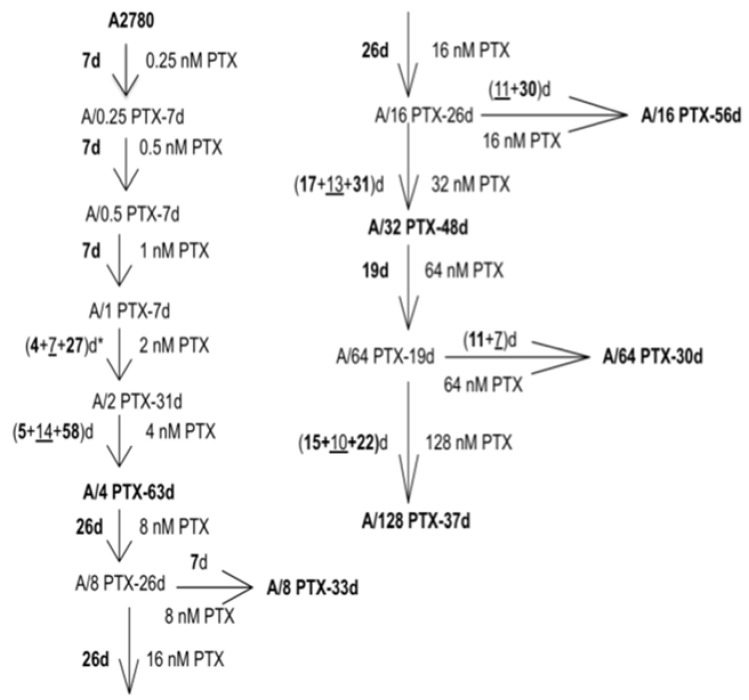
The family tree of A2780 cell sublines with acquired resistance to paclitaxel (PTX). The A2780 parental cell line and six sublines, chosen for further studies, are bolded. The schedule of PTX treatment for each cell line is described as, e.g., (**4** + 7 + **27**)d*—4 days of 2 nM PTX treatment, 7 days of recovery, and 27 days of 2 nM PTX retreatment; a total of 31 days of drug treatment were needed for cell adaptation to 2 nM PTX concentration.

**Figure 2 ijms-21-09218-f002:**
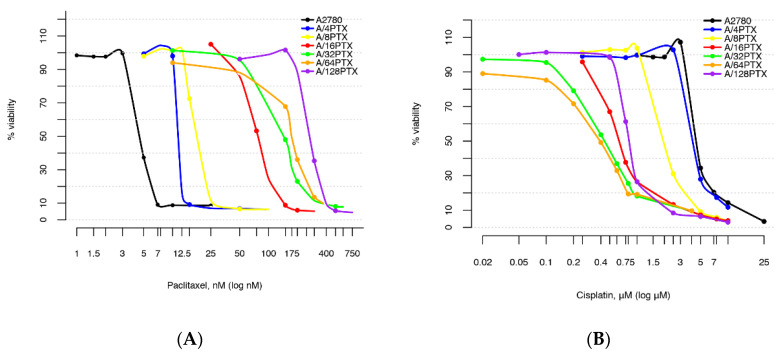
Dose-response curves to PTX and cisplatin (CDDP) for the parental A2780 cell line and six sublines with PTX-induced inverse resistance to PTX and CDDP. (**A**) Resistance to PTX systematically elevates with increasing tolerated doses of PTX. (**B**) Resistance to CDDP declines with increasing tolerated doses of PTX.

**Figure 3 ijms-21-09218-f003:**
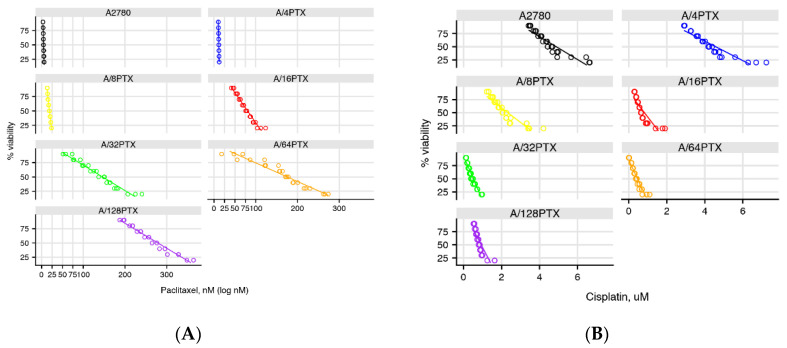
Dose-response relationship approximation to PTX (**A**) and CDDP (**B**) estimated by linear mixed models (LMM).

**Figure 4 ijms-21-09218-f004:**
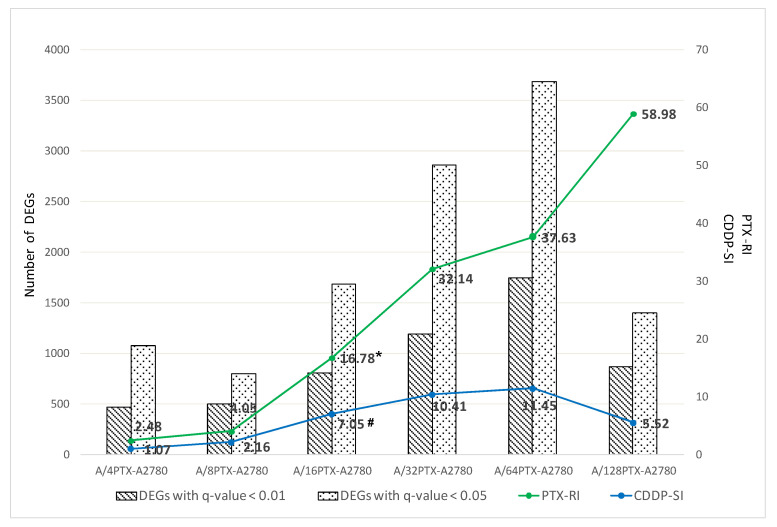
The differences in the number of differentially expressed genes (DEGs) (with *q*-value < 0.01 and *q*-value < 0.05) following the changing differences in PTX resistance/CDDP sensitivity between parental A2780 cells and subsequent sublines with PTX-induced inverse resistance to PTX and CDDP (Table 1 and Appendix A). * RI: Resistance index for PTX is a ratio of IC_50(120h)_ of the resistant cell subline to IC_50(120h)_ of the A2780 cell line, e.g., the A/16PTX cell line PTX-RI was calculated as follows: 77.54/4.62 = 16.78. # SI: CDDP sensitivity index for CDDP is a ratio of IC_50(120h)_ of the A2780 cell line to IC_50(120h)_ of the sensitive cell subline, e.g., the A/16PTX cell line CDDP-SI was calculated as follows: 4.58/0.65 = 7.05.

**Figure 5 ijms-21-09218-f005:**
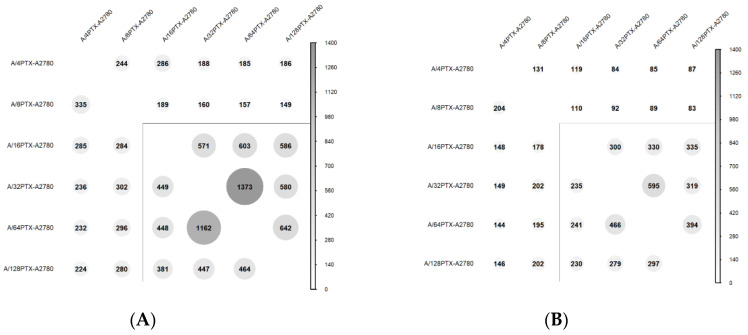
The number of common DEGs between pairs of A2780 cells with six subsequent derived sublines. The circle size and intensity of grey color reflect the number of common DEGs according to the attached scale. (**A**) DEGs with *q*-value < 0.05. Values above the diagonal concern common upregulated genes (total of 6099 genes), whereas values below the diagonal concern common downregulated genes (total of 5825 genes). Two marked lines separate “comparisons with the early sublines” (outside: Total of 1744 upregulated genes vs. 2474 downregulated genes) and “comparisons within the late sublines” (inside: Total of 4355 upregulated genes vs. 3351 downregulated genes). (**B**) DEGs with *q*-value < 0.01. Values above the diagonal concern common upregulated genes (total of 3153 genes), whereas values below the diagonal concern common downregulated genes (total of 3316 genes). Two marked lines separate “comparisons with the early sublines” (outside: Total of 880 upregulated genes vs. 1568 downregulated genes) and “comparisons within the late sublines” (inside: Total of 2273 upregulated genes vs. 1748 downregulated genes).

**Figure 6 ijms-21-09218-f006:**
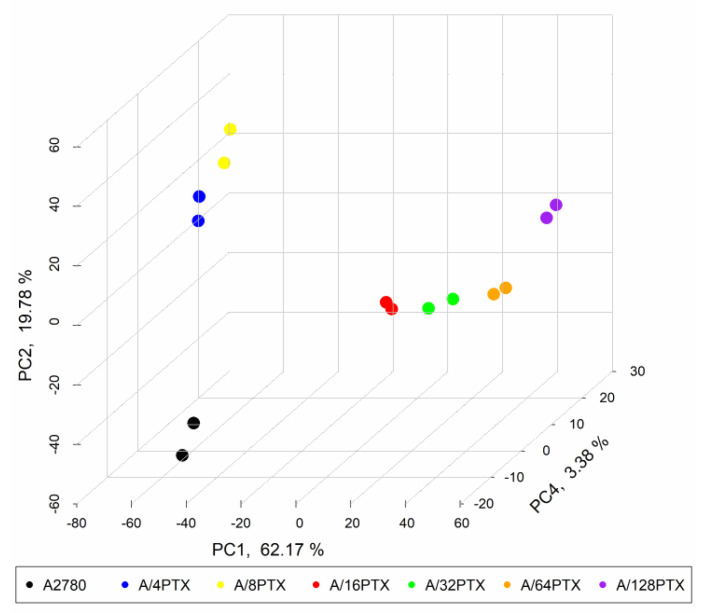
Principle component analysis (PCA) performed on the basis of different gene expressions in A2780 cells and its six derivatives with an inverse resistance to PTX and CDDP. The analysis was performed with the use of 997 of the most significant DEGs (*q*-value < 0.001) independently on two biological replications of each cell line. PC1, PC2, and PC4 components separated seven studied cell lines most clearly.

**Figure 7 ijms-21-09218-f007:**
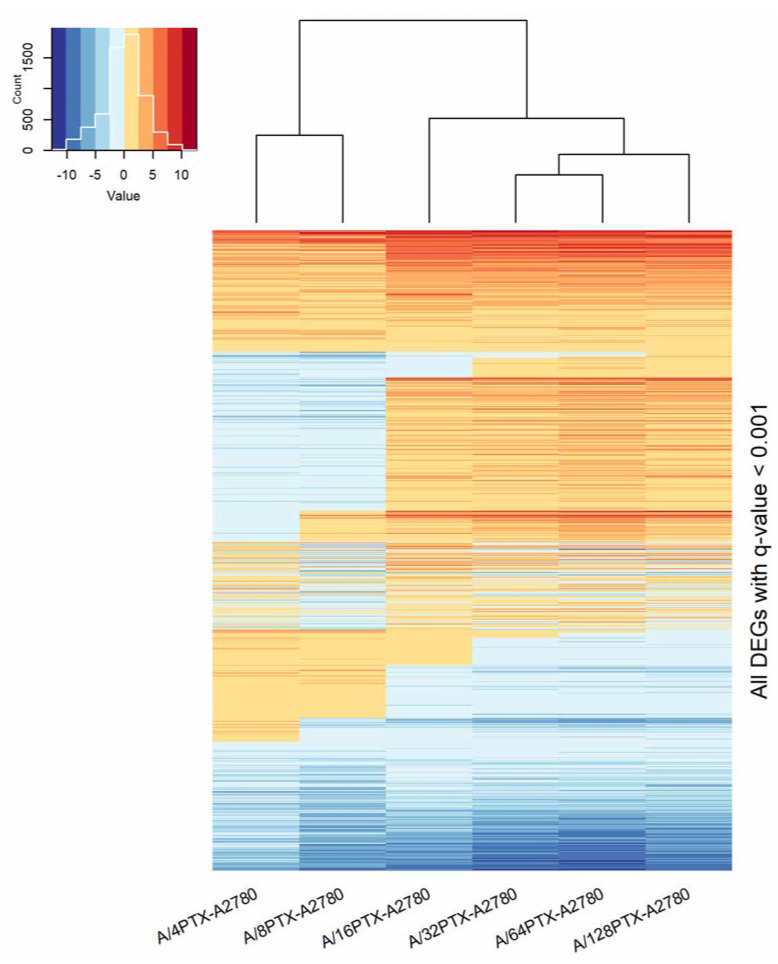
Heat map of gene expression in the development of PTX-induced inverse resistance to PTX and CDDP. The analysis was performed with the use of 997 DEGs with *q*-value < 0.001. The genes are sorted in a way to visualize five expression patterns and thus are not submitted to hierarchical clustering. Downregulated genes are indicated in blue, upregulated genes are indicated in red. The intensity of the color corresponds to the magnitude of the change in expression as depicted in the histogram. The proportion of five DEG expression patterns (consistently upregulated:rising trend:variable expression:descending trend:consistently downregulated) is 190:295:138:174:200.

**Figure 8 ijms-21-09218-f008:**
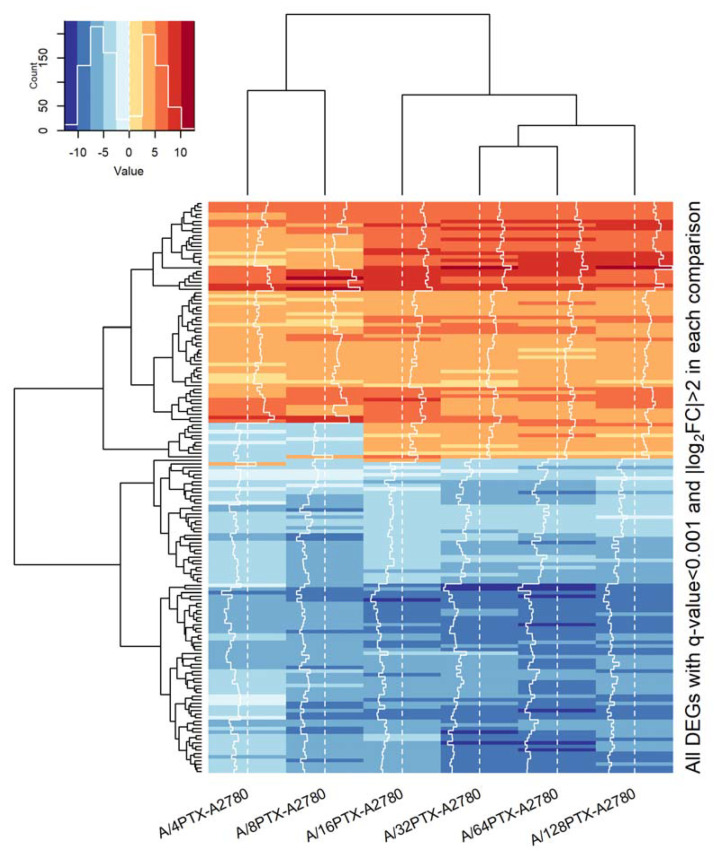
Heat map of gene expression in the development of PTX-induced inverse resistance to PTX and CDDP. The analysis was performed on 160 DEGs with *q*-value < 0.001 and with |log_2_ FC| > 2 in each pair of cell lines comparison. Downregulated genes are indicated in blue, upregulated genes are indicated in red. The intensity of the color corresponds to the magnitude of the change in expression as depicted in the histogram. The proportion of DEG expression patterns (consistently upregulated:rising trend:variable expression:descending trend:consistently downregulated) is 62:10:1:1:86.

**Figure 9 ijms-21-09218-f009:**
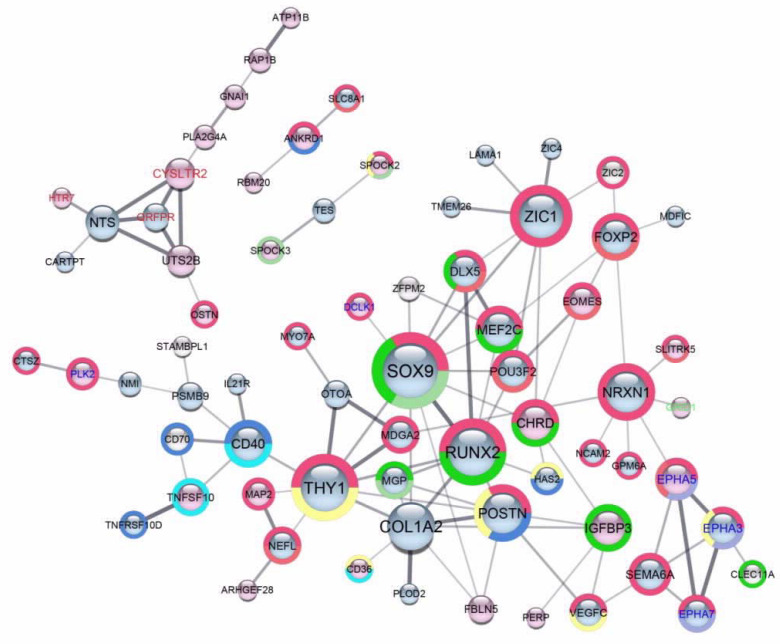
Protein-protein interaction network (PPI) scaffold networks, built on the basis of 160 DEGs (*q*-value < 0.001 and |log_2_ FC| > 2) identified in each one of six A2780-derived sublines compared to the A2780 cell line. Blue nodes represent proteins encoded by downregulated genes (39) and red nodes represent proteins encoded by upregulated genes (26) in six comparisons; silver nodes (5) are proteins encoded by genes with a non-consistent expression pattern. The node size reflects the number of interactions with partners. The edge thickness reflects the strength of protein interaction score. Color split charts around the nodes depict biological processes for which networks are enriched: Two shadows of red: Processes connected with the nervous system; dark green: Ossification; light green: Processes connected with cartilage; yellow: Cell adhesion; dark blue: TNF signaling; blue: Ephrin signaling, and light blue: NFKB1 signaling (Appendix A). Kinases are blue labeled; G protein coupled receptors (GPCRs) are red labeled, and the ion channel is green labeled.

**Table 1 ijms-21-09218-t001:** The values of IC_50(120h)_ for PTX and CDDP in the parental A2780 cell line and six sublines with PTX-developed inverse resistance to PTX and CDDP.

Cell Line	IC_50(120h)_ ± SD for PTX (nM) {RI} ^1^	IC_50(120h)_ ± SD for CDDP (µM) {SI} ^2^
A2780	4.62 ± 0.22 {1}	4.58 ± 0.14 {1}
A/4PTX-63d	11.47 ± 0.07 {2.48}	4.27 ± 0.11 {1.07}
A/8PTX-33d	18.71 ± 0.28 {4.05}	2.12 ± 0.12 {2.16}
A/16PTX-56d	77.54 ± 1.81 {16.78}	0.65 ± 0.01 {7.05}
A/32PTX-48d	148.47 ± 6.82 {32.14}	0.44 ± 0.06 {10.41}
A/64PTX-30d	173.86 ± 4.13 {37.63}	0.40 ± 0.03 {11.45}
A/128PTX-37d	272.47 ± 6.72 {58.98}	0.83 ± 0.04 {5.52}

^1^ RI: Resistance index for PTX is a ratio of IC_50(120h)_ of the resistant cell subline to IC_50(120h)_ of the A2780 cell line, e.g., the A/16PTX cell line PTX-RI was calculated as follows: 77.54/4.62 = 16.78. ^2^ SI: Sensitivity index for CDDP is a ratio of IC_50(120h)_ of the A2780 cell line to IC_50(120h)_ of the sensitive cell subline, e.g., the A/16PTX cell line CDDP-SI was calculated as follows: 4.58/0.65 = 7.05.

**Table 2 ijms-21-09218-t002:** Fold change in resistance to PTX and sensitivity to CDDP, expressed in terms of RI and SI, in consecutive cell sublines. Shadowed codes represent breakthroughs in the development of an inverse resistance to PTX and CDDP.

Pair of Consecutive Cell Lines	PTX-RI ^1^	CDDP-SI ^2^
A2780–A/4PTX	2.48	1.07
A/4PTX–A/8PTX	1.63	2.01
A/8PTX–A/16PTX	4.14	3.26
A/16PTX–A/32PTX	1.91	1.48
A/32PTX–A/64PTX	1.17	1.10
A/64PTX–A/128PTX	1.57	0.48

^1^ PTX-RI is a ratio of IC_50(120h)_ of the child cell line to IC_50(120h)_ of the parental cell line, e.g., the A/8PTX–A/16PTX pair PTX-RI was calculated as follows: 77.54/18.71 = 4.14. ^2^ CDDP-SI is a ratio of IC_50(120h)_ of the parental cell line to IC_50(120h)_ of the child cell line, e.g., the A/8PTX–A/16PTX pair CDDP-SI was calculated as follows: 2.12/0.65 = 3.26.

**Table 3 ijms-21-09218-t003:** Pairwise comparison of the differences in cell line survival according to LMM versus (vs.) Student’s t-test on mean pIC_50_ values. Significant codes: # *p*-value > 0.05; * *p*-value < 0.05; ** *p*-value < 0.01; *** *p*-value < 0.001. Shadowed codes represent results evaluated as not significant in LMM, whereas statistically significant in the t-test.

Pair of Cell Lines	*p*-Value for PTX/CDDP (In LMM)	*p*-Value for PTX/CDDP (In t-Test)
A2780–A/4PTX	*****/#**	*****/***
A2780–A/8PTX	*****/*****	*****/*****
A2780–A/16PTX	*****/*****	*****/*****
A2780–A/32PTX	*****/*****	*****/*****
A2780–A/64PTX	*****/*****	*****/*****
A2780–A/128PTX	*****/*****	*****/*****
A/4PTX–A/8PTX	*****/*****	*****/*****
A/4PTX–A/16PTX	*****/*****	*****/*****
A/4PTX–A/32PTX	*****/*****	*****/*****
A/4PTX–A/64PTX	*****/*****	*****/*****
A/4PTX–A/128PTX	*****/*****	*****/*****
A/8PTX–A/16PTX	*****/*****	*****/*****
A/8PTX–A/32PTX	*****/*****	*****/*****
A/8PTX–A/64PTX	*****/*****	*****/*****
A/8PTX–A/128PTX	*****/*****	*****/*****
A/16PTX–A/32PTX	*****/*****	*****/***
A/16PTX–A/64PTX	*****/*****	*****/****
A/16PTX–A/128PTX	*****/#**	*****/****
A/32PTX–A/64PTX	*****/#**	***/#**
A/32PTX–A/128PTX	**#/***	*****/****
A/64PTX–A/128PTX	****/***	*****/*****

Appropriate 95% confidence intervals (95% CI) of differences between slope parameters (estimates) in LMM and differences between pIC_50_ means in the pairwise comparison of cell lines in the Student’s t-test are depicted in Appendix A, respectively.

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
