# Peer review of "Transcriptome Remodeling in Gradual Development of Inverse Resistance between Paclitaxel and Cisplatin in Ovarian Cancer Cells"

_ijms, 2020, doi:10.3390/ijms21239218_

Round 1
Reviewer 1 Report
Review of “Osteogenesis and Chondrogenesis Associated Genes Are Involved in Gradual Development of Inverse Resistance to Paclitaxel and Cisplatin in Ovarian Cancer Cells” by Szenajch et al.
In this manuscript the authors describe the creation of a series of cell lines derived from the ovarian cancer cell line A2780 by selecting for increasing resistance to paclitaxel (PTX), a drug commonly used to treat ovarian cancers in combination with platinum compounds. They used the analysis of differentially expressed genes (DEGs) between the parental and derived cells to gain insight into mechanisms underlying inverse resistance between PTX and cisplatin (CDDP).
The interdependency of resistance pathways to drugs is an interesting and important phenomenon and insights into the underlying mechanism could be potentially of great clinical relevance. (It should be pointed out that for ovarian cancer chemotherapy taxanes are already usually combined with platinum compounds.) Unfortunately this manuscripts provides few mechanistic insights, yet the current work could be expanded to do so.
Major Points:
The six cell lines were derived from A2780 over a period of around 8 months by selecting cells surviving an increasing drug dose. It is important to realize that – like cell lines in general – A2780 are not a homogenous cell population. During that period a) cells could acquire resistance, b) an ab initio resistant subpopulation could be selected for, c) besides advantageous mutations and pathway activations, also passenger mutations and non-detrimental pathway activations can occur, or d) (likely) a combination of these effects happens. It is thus not surprising that, overall, more than a thousand DEGs were identified. Despite an extensive comparative analysis, it is thus impossible to determine which gene changes contributed to a) resistance to PTX, or b) CDDP sensitivity – or even whether a) and b) are correlated.
Possible approaches could be a comparison of different resistant cell lines derived (by independent selection) from the same parental cell line to identify common pathways changes. These would then have to be validated by interference (knockdown, knockout, or chemical inhibition) to have a role in the drug response. An interesting question is also whether immediate exposure of A2780 to the maximum PTX dose (128 nM) would select for a small subpopulation of resistant cells and how such derived cells would compare to parental and stepwise selected cells. In this regard, it should be noted that drug responses of a population usually do not follow a linear dose-response curve (as assumed in Fig. 3) but take a sigmoidal shape.
In general, the manuscript is very lengthy. Particularly the dragging discussion describes pathways that were never confirmed to have an actual impact on resistance. Correlation of gene expression with resistance is portrayed as causative.
The language could be improved in several instances and some paragraphs are cumbersome.
Suggestion: The authors spend considerable efforts and time in making the cell lines. They are advised to characterize these biologically (cell cycle time, cell death mechanisms after drug exposure etc.) that could provide mechanistic insights and – maybe – hints on what DEGs might be important. A validation of a causal relation between a pathway (PLK2?) and PTX resistance / CDDP sensitivity would greatly improve this currently premature manuscript.
Author Response
Response to Reviewer 1 Comments
Point 1: The six cell lines were derived from A2780 over a period of around 8 months by selecting cells surviving an increasing drug dose. It is important to realize that – like cell lines in general – A2780 are not a homogenous cell population. During that period a) cells could acquire resistance, b) an ab initio resistant subpopulation could be selected for, c) besides advantageous mutations and pathway activations, also passenger mutations and non-detrimental pathway activations can occur, or d) (likely) a combination of these effects happens. It is thus not surprising that, overall, more than a thousand DEGs were identified. Despite an extensive comparative analysis, it is thus impossible to determine which gene changes contributed to a) resistance to PTX, or b) CDDP sensitivity – or even whether a) and b) are correlated.
Response 1: Thank you very much for your pertinent comment. We considered this point of view in our manuscript (lines 873-881).
Point 2: Possible approaches could be a comparison of different resistant cell lines derived (by independent selection) from the same parental cell line to identify common pathways changes. These would then have to be validated by interference (knockdown, knockout, or chemical inhibition) to have a role in the drug response. An interesting question is also whether immediate exposure of A2780 to the maximum PTX dose (128 nM) would select for a small subpopulation of resistant cells and how such derived cells would compare to parental and stepwise selected cells. In this regard, it should be noted that drug responses of a population usually do not follow a linear dose-response curve (as assumed in Fig. 3) but take a sigmoidal shape.
Response 2: The idea of comparing stepwise with independently selected cells is very interesting. However, on the basis of our experience (please look at Fig. 1 and its description in our manuscript (subsection 2.1)), we are afraid that independent (started from parental A2780 cell line treatment) developing of resistant sublines will be easy only for low PTX concentrations (0.5–1 nM), difficult for 2 nM PTX and very difficult or almost impossible for higher concentrations (4-128 PTX). According to our several year experience with drug treatment of cell culture, we are almost sure that higher concentrations of PTX would kill all or almost all sensitive A2780 cells very quickly and selection of resistant subpopulation would be rather impossible.
Proposed validation of pathways changes by mechanistic studies are essential for elucidation of their actual role in resistance and such approach will be the next step in our research. In the present manuscript we focus on evolutionary changes in gene expression between the isogenic cell lines with stepwise generated drug resistance. These changes point out the directions of further studies.
We agree with the statement that the drug response of a population usually does not follow a linear dose-response curve. However, a difference in cell survival is one of the most obvious manifestation of a population of cells submitted to treatment, and should be measured as an important element of drug response. That is why we were searching methods that would allow us to verify which lines significantly differ in cell survival. The linear model that was applied is the method that allowed us to evaluate whether the speed of changes in survival between cell lines is relevant. In relation to the Reviewer’s comment, we changed the description of Fig. 3 from “dose-response curve” to “dose-response relationship approximation” to underline this fact (line 271). Consistently, we made appropriate changes in the text (lines 268-270 and 308).
Point 3: In general, the manuscript is very lengthy. Particularly the dragging discussion describes pathways that were never confirmed to have an actual impact on resistance. Correlation of gene expression with resistance is portrayed as causative.
Response 3: We tried to shorten and simplify the discussion.
We very carefully verified the cited literature in respect of confirmed or putative impact of proteins encoded by scaffold DEGs on drug resistance and tumor promotion and we referred to these issues in our discussion (SOX9 and RUNX2 – lines 635-642; POSTN – lines 650-652; MGP – lines 652-654; SPOCK2 and SPOCK3 – lines 658-663; VEGFC – lines 664-666; THY1 – lines 712-715; ZIC1 – lines 718-720; TNFSF10 and TNFRSF10D – lines 724-731, 745-749; CD40 – lines 760-763; CD70 - lines 766-769; EPHA3 – line 791; and PLK2 – lines 802-827). The found correlation of expression of genes associated with osteo- and chondrogenesis with resistance development is one of the most interesting results of our study. It is also new, because up to date the osteomimicry connected phenomena have been associated with bone metastases, not with resistance acquiring.
We agree that without functional studies we cannot interpret the correlation of deregulated gene expression with resistance as a simple cause-result relation and we highlighted this in several fragments of text (lines: 38-40, 127-129, 559-561, 882-883 and 1087-1089).
Point 4: The language could be improved in several instances and some paragraphs are cumbersome.
Response 4: We intend to take advantage of the MDPI English editing service as soon as the manuscript is approved.
Point 5: The authors spend considerable efforts and time in making the cell lines. They are advised to characterize these biologically (cell cycle time, cell death mechanisms after drug exposure etc.) that could provide mechanistic insights and – maybe – hints on what DEGs might be important. A validation of a causal relation between a pathway (PLK2?) and PTX resistance / CDDP sensitivity would greatly improve this currently premature manuscript.
Response 5: We plan to perform detailed biological characterization of cells and functional studies as well. The possible role of PLK2 in development of inverse resistance to PTX and CDDP and observed abrogation of osteomimic phenotype accompanying the drug resistance development seem to be the most interesting aims of our future research. The aim of this manuscript was to present our cellular model and research possibilities which it provides for resistance studies. This is especially important for our team, as our model will be soon available in the international cell culture collection.
Summarizing, we thank the Reviewer 1 for the valuable comments. We believe that they helped to improve our manuscript.
Reviewer 2 Report
The authors present a well-written manuscript outlining the establishment of a highly relevant model of drug resistance to investigate differential gene expression.
Author Response
Response to Reviewer 2 Comments
Point 1: The authors present a well-written manuscript outlining the establishment of a highly relevant model of drug resistance to investigate differential gene expression.
Response 1: We thank the Reviewer 2 for the positive review.
We intend to take advantage of the MDPI English editing service as soon as the manuscript is approved.
Reviewer 3 Report
Understanding molecular mechanism of drug resistant is generally very important to improve cancer patients’ outcomes, especially ovarian cancer as the recurrence is quite common after standard chemotherapy.
This basic science research showed the view of altered gene expression which contribute drug resistance development of ovarian cancer.
Their methodology is novel and well described in the article.
Among their findings, the abrogation of osteomimic phenotype in ovarian cancer cells during the development of inverse resistance to PTX and CDDP is novel and intriguing. I hope they expand their research and apply their findings to translational research in clinical setting.
Author Response
Response to Reviewer 3 Comments
Point 1: Understanding molecular mechanism of drug resistant is generally very important to improve cancer patients’ outcomes, especially ovarian cancer as the recurrence is quite common after standard chemotherapy.
This basic science research showed the view of altered gene expression which contribute drug resistance development of ovarian cancer.
Their methodology is novel and well described in the article.
Among their findings, the abrogation of osteomimic phenotype in ovarian cancer cells during the development of inverse resistance to PTX and CDDP is novel and intriguing. I hope they expand their research and apply their findings to translational research in clinical setting.
Response 1: We thank you the Reviewer 3 for positive review. It is especially nice for us that our findings concerning abrogation of osteomimic phenotype during the resistance development were appreciated. We consider these results as the most interesting in our work and plan to expand our future research in this direction.
Round 2
Reviewer 1 Report
The authors have made only minor revision to the text without addressing the reviewer's main concerns. In their reply the authors pointed out that the novelty in their manuscript is the creation paclitaxel (Taxol) resistant cell lines. It should be pointed out that Taxol resistant cells have been established previously, including A2780/Taxol (Wang et al., PMID: 24175763) or Blachandran et al. (PMID: 14654788). The title still implies a causal relationship between "Osteogenesis and Chondrogenesis Associated Genes and Development of Inverse Resistance to Paclitaxel and Cisplatin in Ovarian Cancer Cells", when the data do not support such a relation.
Author Response
Response to Reviewer 1 Comments – Round 2
Point 1: The authors have made only minor revision to the text without addressing the reviewer's main concerns.
Response 1: In the first round of review we tried to address all Reviewer 1 concerns and consider the suggestions in our manuscript. We would like here to summarize briefly our 1st revision:
- We into the Discussion the Reviewer’s point of view on gene changes and their contribution in developing drug resistance
- We discuss the proposed by Reviewer 1 approach consisted in comparison of stepwise and independently selected drug resistant cells. We explained our idea of using the linear model to measure drug response and modified the description of Fig.3.
- We tried to shorten and simplify Discussion. We also carefully verified the cited literature to support the putative impact of proteins encoded by scaffold DEGs on drug resistance and tumor promotion. Cancer- and drug resistance-related genes and pathways were given top priority.
In several fragments of the text we highlighted that future functional studies are needed to assess the casual relationship between gene changes and drug response.
In the 2nd round of review we decided to weaken more the tone of our argumentation to avoid over-interpretation of the obtained results. We also tried to make the Discussion more concise.
Point 2: In their reply the authors pointed out that the novelty in their manuscript is the creation paclitaxel (Taxol) resistant cell lines. It should be pointed out that Taxol resistant cells have been established previously, including A2780/Taxol (Wang et al., PMID: 24175763) or Blachandran et al. (PMID: 14654788).
Response 2: According to our declaration in the Abstract “In this study we proposed a new, transcriptome-focused approach, utilizing a model of isogenic cancer cell lines with gradually changing resistance.” (lines 22-23), we do not treat the process of development of PTX resistant cell lines itself as the novelty of our manuscript. In the Introduction, we widely refer to other studies on cell models of acquired resistance, in particular to PTX and CDDP. The first cell lines with in vitro induced drug resistance were developed more than 30 years ago (lines 81-82) and nowadays there is no way to cite all valuable papers which have appeared in this field. Also in the Discussion we shortly summarized other authors’ efforts and results in resistance induction in different ovarian cancer cell lines. In the round 2 we added to these references one article suggested by the Reviewer (Wang et al., PMID: 24175763) as a reference No 28 (lines 596-598). The second paper listed by the Reviewer 1 (Blachandran et al. PMID: 14654788), though valuable, is beyond the main context of our manuscript, therefore we did not cite it.
Although not novel, the developed inverse resistance to PTX and CDDP, being in line with the most common clinical response, is the advantage of our model (Discussion, lines 605-610). The real novelty of our studies is the approach to created model – using NGS technology we were able to keep track of trends in gene expression and proposed to study the phenomenon of resistance development as a process of evolution, reflected by transcriptome remodeling. Profound analysis of transcriptome enabled us to select the genes encoding proteins which built the scaffold of this process as well as the genes, which expression was changed only at the some stages of resistance acquiring (e.g. only at the early or late stages) (Abstract, lines 31-34).
Point 3: The title still implies a causal relationship between "Osteogenesis and Chondrogenesis Associated Genes and Development of Inverse Resistance to Paclitaxel and Cisplatin in Ovarian Cancer Cells", when the data do not support such a relation.
Response 3: We changed the title for the following one: ”Transcriptome Remodeling in Gradual Development of Inverse Resistance between Paclitaxel and Cisplatin in Ovarian Cancer Cells”.
Summarizing, we introduced changes in the Title, the Introduction, the Discussion and the Conclusions. We thank the Reviewer 1 for the valuable remarks. We believe that they helped to improve our manuscript.